# Isotropic Nature of the Metallic Kagome Ferromagnet Fe$_3$Sn$_2$ at High Temperatures

Rebecca L. Dally [1,*], Daniel Phelan [2], Nicholas Bishop [3], Nirmal J. Ghimire [3,4] and Jeffrey W. Lynn [1]

[1] NIST Center for Neutron Research, National Institute of Standards and Technology, Gaithersburg, MD 20899-6102, USA; jeffrey.lynn@nist.gov
[2] Materials Science Division, Argonne National Laboratory, Lemont, IL 60439, USA; dphelan@anl.gov
[3] Department of Physics and Astronomy, George Mason University, Fairfax, VA 22030, USA; nbishop3@masonlive.gmu.edu (N.B.); nghimire@gmu.edu (N.J.G.)
[4] Quantum Science and Engineering Center, George Mason University, Fairfax, VA 22030, USA
[*] Correspondence: rebecca.dally@nist.gov

**Abstract:** Anisotropy and competing exchange interactions have emerged as two central ingredients needed for centrosymmetric materials to exhibit topological spin textures. Fe$_3$Sn$_2$ is thought to have these ingredients as well, as it has recently been discovered to host room temperature skyrmionic bubbles with an accompanying topological Hall effect. We present small-angle inelastic neutron scattering measurements that unambiguously show that Fe$_3$Sn$_2$ is an isotropic ferromagnet below $T_C \approx 660$ K to at least 480 K—the lower temperature threshold of our experimental configuration. Fe$_3$Sn$_2$ is known to have competing magnetic exchange interactions, correlated electron behavior, weak magnetocrystalline anisotropy, and lattice (spatial) anisotropy; all of these features are thought to play a role in stabilizing skyrmions in centrosymmetric systems. Our results reveal that at the elevated temperatures measured, there is an absence of significant magnetocrystalline anisotropy and that the system behaves as a nearly ideal isotropic exchange interaction ferromagnet, with a spin stiffness $D(T = 480 \text{ K}) = 168$ meV Å$^2$, which extrapolates to a ground state spin stiffness $D(T = 0 \text{ K}) = 231$ meV Å$^2$.

**Keywords:** inelastic neutron scattering; topological materials; anomalous Hall effect; isotropic ferromagnet; kagome; frustrated magnetism; skyrmion; magnetization

## 1. Introduction

The two-dimensional kagome lattice lends itself to hosting a variety of phenomena depending on the chemical species occupying the network of corner-sharing triangles. For example, the tight-binding model for itinerant electrons leads to an electronic spectrum with a flat band and two Dirac crossings at the symmetry protected $K$ and $K'$ corner points of the hexagonal Brillouin zone. Chemical tuning can drive the Fermi level to meet the Dirac points (a Dirac semimetal) to realize chiral massless charge carriers such as that in graphene [1,2]. The prediction of the flat band—on the extreme opposite from a Dirac band—is the result of destructive interference of Bloch waves from the lattice geometry. Consequently, this nontrivial flat band can exhibit interesting physics such as flat-band ferromagnetism and a finite Chern number. Experimentally, FeSn was shown to host both flat bands and Dirac fermions [3] due to the isolated Fe kagome layers rendering it a nearly perfect realization of 2D kagome physics. Fe$_3$Sn$_2$ is similar in structure, but features isolated breathing kagome bilayers, as shown in Figure 1a–c. Interestingly, the bilayers and breathing structure were still theorized to have a band structure with similar features. Instead of one Dirac crossing at each $K$ and $K'$ point, there are two which are symmetric about each point [4], and the fermions are both spin-polarized due to the breaking of time-reversal symmetry and massive due to the opening of a gap from spin-orbit coupling. The combination of these effects gives rise to a non-zero Berry curvature which is consistent

with the quadratic relationship of the anomalous Hall resistivity with the longitudinal resistivity [5], implying the intrinsic Karplus and Luttinger mechanism [6] is responsible for the large anomalous Hall effect. It was also recently shown that nearly flat bands near the Fermi surface exist, which may contribute to the observed high-temperature ferromagnetism [7].

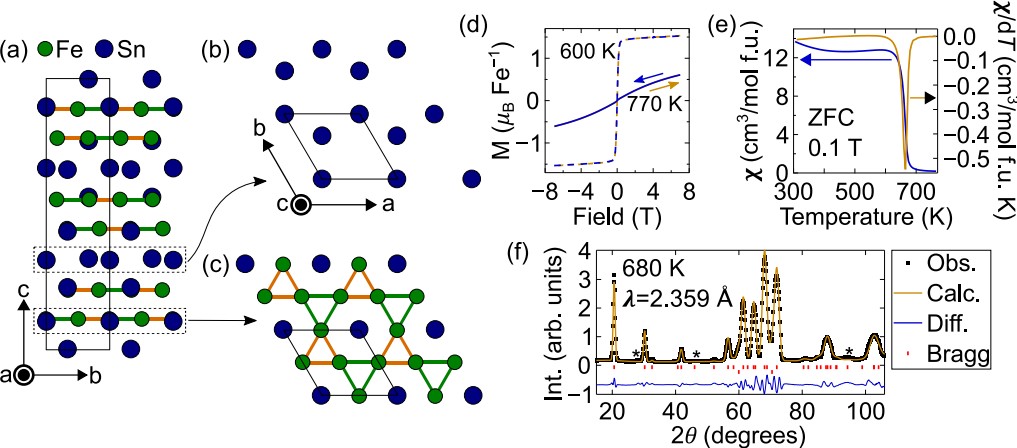

**Figure 1.** Crystal structure and characterization of $Fe_3Sn_2$. The crystallographic space group is $R\bar{3}m$ with reported lattice parameters $a = b = 5.344$ Å and $c = 19.845$ Å [8]. (**a**) View along the **a**-axis. The solid black line represents the unit cell and Fe atoms are shown as the smaller green circles and Sn atoms are shown as the larger blue circles. A single Fe site is offset from any high symmetry position ($x/a = 0.4949$, $y/b = 0.5051$, and $z/c = 0.1134$) leading to two different Fe-Fe bond lengths in the $ab$-plane (the so-called "breathing" kagome). Two Fe-Fe bond lengths are shown, where the shorter bond is in orange, and the longer bond is in green. (**b**) A Sn-only layer viewed along the **c**-axis, where the Sn atoms are arranged on a honeycomb lattice. (**c**) An Fe-Sn layer viewed along the **c**-axis, showing the breathing kagome lattice made up of Fe atoms. The axes labels for (**c**) are the same as in (**b**), and the parallelogram outlined by a solid black line for both panels represents the unit cell. (**d**) Magnetization measurements taken at 770 K (solid lines) and 600 K (dashed lines). The samples show no signs of coercivity as the sweep down in field (blue lines) coincides with the sweep up (orange lines) in field. (**e**) Zero-field cooled (ZFC) magnetic susceptibility measurement taken in a 0.1 T applied magnetic field. The derivative (right axis) clearly shows the ferromagnetic transition at $T_C \approx 665$ K. (**f**) Neutron powder diffraction data taken above the magnetic transition at 680 K. The data demonstrate the structure is consistent with that reported. The upper set of red tic marks denote $Fe_3Sn_2$ Bragg peak positions and the lower set denote Al Bragg peak positions coming from the sample canister. A few small impurity peaks were observed but not identified, and these are marked by the * symbol.

Metallic kagome ferromagnets clearly exhibit elegant physics; however, they are elusive with only two reported: $Co_3Sn_2S_2$, a semimetal [9], and the aforementioned $Fe_3Sn_2$. As alluded to thus far, the electronic structure has signatures of non-trivial topology, but recently, $Fe_3Sn_2$ has garnered growing attention for the discovery of topologically non-trivial spin textures. The observation of room temperature skyrmion bubbles [10] quickly led to reports of nanostructured skyrmionic devices [11–14] and studies of the associated properties such as the topological Hall effect [15–17] and skyrmion thermopower [18]. The space group of $Fe_3Sn_2$ is the centrosymmetric $R\bar{3}m$, meaning the mechanism for skyrmion bubble formation is not due to the conventional breaking of crystalline inversion symmetry with Dzyaloshinshkii-Moriya interactions found in conventional B20 skyrmion systems. Instead, topological magnetic structures in centrosymmetric systems are due to the presence of anisotropy and/or frustration. The underlying source of each which is needed to stabilize skyrmions has become widely studied in recent years. One common model is the triangular lattice with frustrated Heisenberg antiferromagnetic exchange

interactions. [19] Although the magnetic frustration alone was shown to lead to a skyrmion phase, either lattice and/or spin anisotropy [20], particularly easy-axis anisotropy [21,22], was shown to be helpful in stabilizing the skyrmions.

From a frustrated magnetism perspective, $Fe_3Sn_2$ has been of interest for quite some time. Original neutron powder diffraction measurements indicated collinear ferromagnetic order below the onset of magnetism at $T_C \approx 660$ K with moments oriented along the *c*-axis [23]. A spin-reorientation transition to the *ab*-plane starting below 250 K was identified and later measurements implied a slightly non-collinear structure was more likely starting below 300 K [24] and that the spin-reorientation transition was actually first-order in nature and occurs at $\approx$150 K [25,26]. The non-collinearity is thought to be due to frustrated magnetic exchange, and would also explain the large anomalous Hall effect [5,27] and possibly some of the temperature regimes where the topological Hall effect is observed if the scaler spin chirality is finite. Bulk magnetic measurements have also shown $Fe_3Sn_2$ to be an extremely soft ferromagnet at all temperatures with no coercivity, implying any easy-axis magnetic anisotropy must be very weak.

Here, we present our small-angle inelastic neutron scattering study of the magnetic excitations in $Fe_3Sn_2$ between 480 K and 660 K and unambiguously show that no significant spin wave gap is observed within experimental uncertainties between these temperatures. Below 480 K, the spin stiffness parameter, $D(T)$, becomes too large and the spin wave full-width-at-half-maximum in energy, $\Gamma(q)$, has narrowed to the point that the excitations move outside our measurement window. However, our results show that down to at least 480 K, $Fe_3Sn_2$ behaves as an ideal isotropic ferromagnet, and any onset of significant magnetic anisotropy that may contribute to the topological spin textures must develop below this point.

## 2. Materials and Methods

Polycrystalline samples of $Fe_3Sn_2$ were synthesized by solid state reaction. Stoichiometric amounts of Fe powder (Alfa Aesar 99+%) and Sn powder (Alfa Aesar 99.995%) were mixed and pelletized. The pellet was sealed in a fused silica ampoule under vacuum. The sealed ampoule was heated to 800 °C at the rate of 1 °C/hour and was kept at 800 °C for 1 week. After 1 week, the ampule at 800 °C was quenched into ice water. The pellet was reground, re-pelletized, and sealed into the fused silica ampoule under vacuum and was annealed at 800 °C for 1 week.

Magnetic measurements were performed on a piece of pressed pellet of $Fe_3Sn_2$ powder employing a Quantum Design MPMS3 magnetometer with an oven heater stick between 300 K and 756 K.

Neutron powder diffraction (NPD) measurements were taken using the triple-axis spectrometer, BT-7, at the NIST Center for Neutron Research [28]. A 17 g sample of polycrystalline $Fe_3Sn_2$ was sealed in a cylindrical aluminum canister, which was mounted inside a closed cycle refrigerator. Data were collected in two-axis mode using a position sensitive detector and wavelength of 2.359 Å. Söller collimators of $50' - 40'R$ were used before and after the sample, respectively (where *R* indicates radial), and pyrolytic graphite (PG) filters were employed both in the reactor beam and after the sample to suppress higher order wavelength contributions. Data were refined using the Rietveld method and the program, FullProf [29].

Inelastic neutron scattering data were also taken using BT-7 and the same 17 g sample as in NPD. Two different small-angle inelastic neutron scattering configurations were used in order to obtain data over a wider temperature range. For higher temperatures (630 K to 660 K), PG(002) monochromator crystals with vertical focusing and PG(002) analzyer crystals were used, and constant-Q scans were taken with a fixed incident energy of 13.7 meV. Söller collimators of $10' - 10' - 10' - 25'$ were used before and after the monochromator and before and after the analyzer, and the reactor beam PG filter was once again employed. The vertical resolution was measured using a graphite crystal and found to be 0.16 Å$^{-1}$. For lower temperatures (480 K to 610 K), PG(004) monochromator crystals

with vertical focusing and PG(004) analzyer crystals were used, and constant-*Q* scans were taken with a fixed incident energy of 35 meV. A velocity selector in the reactor beam was employed to suppress higher and lower order wavelengths. The same collimations as the $E_i = 13.7$ meV experiment were used and the vertical resolution at this higher energy was found to be 0.24 Å$^{-1}$. The same scans taken at high temperatures were also taken at much lower temperatures (300 K for the $E_i = 13.7$ meV experiment and 250 K for the $E_i = 35$ meV experiment), and these data were used for background subtraction.

In the small-angle inelastic neutron scattering configuration, spin waves are probed in the long-wavelength (i.e., small-*q*) limit, and the dispersion for a ferromagnet is

$$\hbar\omega(q) = \Delta + D(T)q^2, \tag{1}$$

where $\Delta$ is any anisotropy gap and $D(T)$ is the spin wave stiffness which in mean field theory is proportional to the magnetization. The kinematic constraints for the scattering severely restrict the range of energy transfers accessible, so that the spin waves can only be observed if there is little to no anisotropy gap. The point in reciprocal space where the spin waves are being probed can be viewed by the schematic in Figure 2a. Here, a parabolic dispersion about $Q = 0$ and energy transfer, $E = 0$, is shown. About this point, the dispersion of an isotropic ferromagnet powder sample is identical to that of a single crystal. Similar experiments on amorphous alloys [30] and powder samples of manganites [31] have been widely used to establish their isotropic nature.

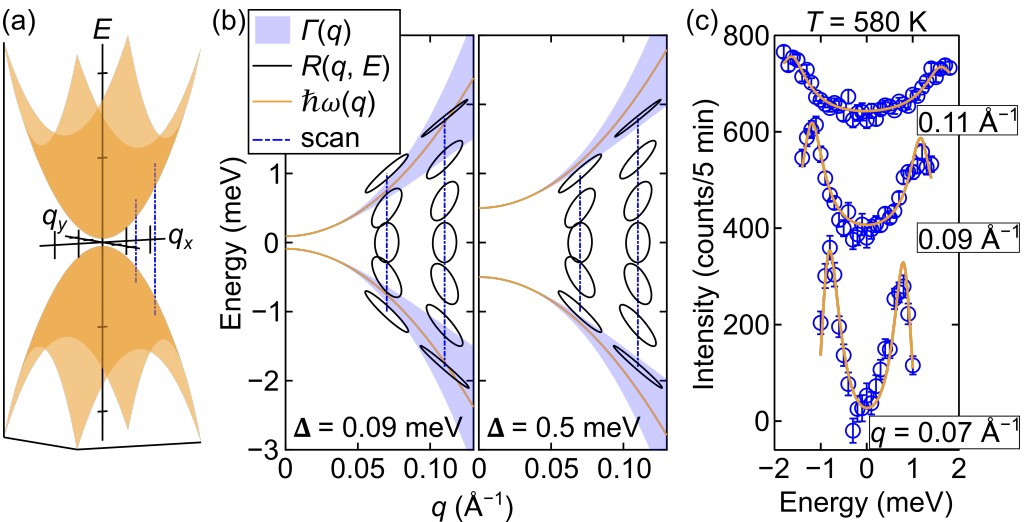

**Figure 2.** (**a**) A three-dimensional schematic of an isotropic parabolic spin wave dispersion near $Q = 0$ and energy transfer, $E = 0$. The dispersion is shown as the orange surface, and dashed blue lines show the direction of constant-*Q* scans cutting through the dispersion surface along *E*. (**b**) A two-dimensional schematic demonstrating how the experiment captures the intensity from the spin wave excitations. The orange solid line represents the dispersion, $\hbar\omega(q) = \Delta + D(T)q^2$, and the surrounding blue surface represents the full-width-at-half-maximum of the spread in energy of the dispersion, $\Gamma(q)$, due to thermally induced magnon-magnon interactions. Dashed blue lines are examples of constant-*Q* scans made in the experiment. Overlayed on these lines are the instrumental resolution ellipses, $R(Q, E)$, along $Q = 0.07$ Å$^{-1}$ and 0.11 Å$^{-1}$. The left panel uses the refined values for the dispersion from the actual data at $T = 580$ K in the $E_i = 35$ meV experiment. The right panel uses the same parameters, but increased the gap to be 0.5 meV in order to demonstrate the sensitivity of the technique to the size of the gap. Here, the scans performed during the experiment wouldn't be able to reach the signal of the spin waves. (**c**) The actual data at $T = 580$ K in the $E_i = 35$ meV experiment. The solid orange lines are the refined fits to the data, shown as blue circles. The three constant-*Q* scans are vertically offset from one another for clarity.

One advantage of studying the spin waves in the long-wavelength limit (i.e., about $Q = 0$) is that the instrumental resolution is focused on both the energy gain and energy loss side, unlike with a Bragg point where there is a focused and de-focused side. More details on the resolution function in the small-angle limit can be found in Ref. [32] The intensity detected in a neutron scattering experiment represents a convolution of the instrumental resolution, $R(Q, E)$, and the scattering function, $S(q, \hbar\omega)$, making it necessary to include the convolution when analyzing the data. Using the Cooper-Nathans approximation for the resolution, we used the program ResLib [33] to fit each set of data (where a set of data consists of all the constant-$Q$ scans at a single temperature) to the scattering function,

$$S(q, \hbar\omega) \propto F(q, \hbar\omega) \frac{\hbar\omega}{1 - e^{-\hbar\omega/k_B T}}, \tag{2}$$

where $F(q, \hbar\omega)$ is the spectral weight function, $k_B$ is the Boltzmann constant, and $T$ is the temperature. At finite temperatures, magnon-magnon interactions lead to damping effects in energy for the spin waves. For Heisenberg ferromagnets below $T_C$, and in the energy regime $\hbar\omega \ll k_B T$, the excitation width in energy has been calculated [34] as

$$\Gamma(q) \propto q^4 T^2 \left\{ \frac{1}{6} \ln^2\left(\frac{k_B T}{\hbar\omega}\right) + \frac{5}{9} \ln\left(\frac{k_B T}{\hbar\omega}\right) - 0.05 \right\}. \tag{3}$$

The shape of the broadening in energy is approximated using a Lorentzian function as the spectral weight function, $F(q, \hbar\omega)$, centered about $\hbar\omega(q)$, with $\Gamma(q)$ as the full-width-at-half maximum.

## 3. Results

### 3.1. Characterization

Figure 1d shows the magnetization both above (770 K) and below (600 K) the ferromagnetic transition temperature. No coercivity was observed for either temperature, meaning $Fe_3Sn_2$ is a soft ferromagnet. Figure 1e shows the magnetic susceptibility as a function of temperature at an applied magnetic field of 0.1 T. The derivative of the susceptibility with respect to temperature shows the ferromagnetic transition to be $\approx 665$ K, which is consistent with previous reports that show the Curie temperature to vary anywhere between 640 K and 660 K [5,16,24].

The observed NPD profile and Rietveld refined fit are shown in Figure 1f. The data were taken at 680 K and confirm the structure to be $Fe_3Sn_2$ with refined lattice parameters of $a = b = 5.3787 \pm 0.0004$ Å and $c = 19.863 \pm 0.002$ Å. The refined atomic positions for Fe are $x/a = 0.4940 \pm 0.0004$, $y/b = 0.5060 \pm 0.0004$, and $z/c = 0.1132 \pm 0002$. The Sn positions are $z/c = 0.1039 \pm 0.0007$ and $z/c = 0.3318 \pm 0.0007$ for the Sn1 and Sn2 sites, respectively. The isotropic thermal parameters ($B$) were refined to $0.8 \pm 0.1$ Å$^2$, $4.1 \pm 0.5$ Å$^2$, and $2.6 \pm 0.4$ Å$^2$ for the Fe, Sn1, and Sn2 sites, respectively. There were three small impurity peaks in the pattern that were unable to be identified. They are marked with an * in Figure 1f.

### 3.2. Inelastic Neutron Scattering

We first demonstrate the sensitivity of the small-angle inelastic scattering configuration to the size of the gap in order to discern between isotropic and anisotropic ferromagnets. A schematic of a dispersion following Equation (1) near $\mathbf{Q} = 0$ is shown in Figure 2a. The neutron scattering plane is defined by two arbitrary orthogonal vectors, $\mathbf{q}_x$ and $\mathbf{q}_y$, and constant-$Q$ cuts are shown as blue dashed lines to show how the experimental scans can cut through the dispersion along energy, $E$.

Each temperature set of constant-$Q$ scans was fit globally to obtain the spin wave parameters, and the parameters for $T = 580$ K were used to create Figure 2b. The spin stiffness parameter was found to be $D(T = 580$ K$) = 135 \pm 3$ meV Å$^2$ and the gap, $\Delta = 0.09 \pm 0.02$ meV, where the uncertainties throughout represent one standard deviation due to statistical counting. These parameters were used to create the solid orange line

representing $\hbar\omega(q)$. We note that even for an ideal isotropic spin system a small dipolar gap is expected due to ferromagnetic magnetization. The full-width-at-half-maximum of the spin waves in energy, $\Gamma(q)$, is shown as the shaded blue region following Equation (3), and the instrumental resolution, $R(q, E)$, is shown as black ellipses. The maxima and minima of the ellipses along energy represent the allowed scan region which satisfies the required conservation of momentum and energy represented by the scattering triangle (i.e., scanning farther in energy is not possible). Two representative constant-$Q$ scans are shown as dashed blue lines at 0.07 Å$^{-1}$ and 0.11 Å$^{-1}$, and with the small gap of 0.09 meV, the center of the instrumental resolution ellipses for both scans is able to pass over the peak of the dispersion on the energy gain and loss sides. This is not possible if the gap is increased to 0.5 meV, as shown in the right panel of Figure 2b (all other parameters from the $T = 580$ K fit were fixed). The actual data from $T = 580$ K are shown in Figure 2c as blue circles and the fits are shown as solid orange lines.

The spin stiffness parameter, $D(T)$, was extracted from the fits for each temperature and is shown in Figure 3. The dashed line is a power law fit to the data: $D(T) = D_0 \left( \frac{T_C - T}{T_C} \right)^{\nu - \beta}$, where $D_0 = 271 \pm 9$ meV Å$^2$, $T_C = 662.4 \pm 0.8$ K, and $\nu - \beta = 0.34 \pm 0.02$. The solid line is a fit to the Dyson formalism of two spin-wave interactions in a Heisenberg ferromagnet, where $D(T) = D_0 \left[ 1 - A \left( \frac{k_B T}{4\pi D_0} \right)^{5/2} \zeta\left(\frac{5}{2}\right) \right]$, $\zeta\left(\frac{5}{2}\right)$ is the Riemann integral and $A$ is a constant proportional to the interaction range [35]. The $T^{5/2}$ temperature dependence is not valid near the critical regime, and only the lowest four temperatures were used in the fit, resulting in $D_0 = 231 \pm 7$ meV Å$^2$.

The gap was not found to have any meaningful temperature dependence, ranging between 0.06 meV and 0.09 meV, and was the same within plus or minus one standard deviation. It should also be noted that the instrumental resolution in energy for the scans taken is on the order of these values (see Figure 2b), meaning the exact fitted value for the gap is not well-defined. For example, in the $E_i = 13.7$ meV experiment, the resolution in energy at $Q = 0.07$ Å$^{-1}$ and $E = 0$ meV is 0.29 meV, and at $E = 0.6$ meV the resolution is 0.07 meV.

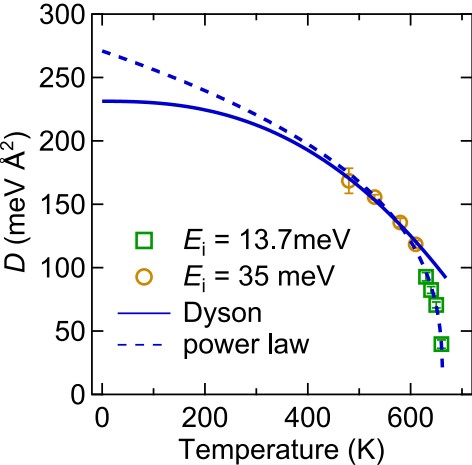

**Figure 3.** The temperature dependence of the spin wave stiffness parameter, $D(T)$. Data from both the $E_i = 13.7$ meV and $E_i = 35$ meV experiments were included in the power law fit, $D(T) = D_0 \left( \frac{T_C - T}{T_C} \right)^{\nu - \beta}$, shown as the dashed line. Only the lowest four temperatures were used in the Dyson fit, $D(T) = D_0 \left[ 1 - A \left( \frac{k_B T}{4\pi D_0} \right)^{5/2} \zeta\left(\frac{5}{2}\right) \right]$, shown as the solid line.

## 4. Discussion

The temperature renormalization of the spin stiffness for Heisenberg ferromagnets is expected to follow a power law on approach to $T_C$ with the critical exponents $\nu - \beta = 0.34$ [36], which is the exponent found in this study, further showing that magnetically, $Fe_3Sn_2$ is a

typical exchange ferromagnet at elevated temperatures. In fact, the $\nu - \beta$ exponent from the power law fit is strikingly similar to those for elemental Fe [37] ($\nu - \beta = 0.37$) and Ni [38] ($\nu - \beta = 0.39$), in addition to the amorphous iron magnets already mentioned in Ref. [30], using the same technique. We remark that the $T_C$ of 631 K for Ni is comparable to $Fe_3Sn_2$, but the spin stiffness of $D = 550$ meV $Å^2$ in the ground state of this itinerant magnet is much larger [39]. However, the extrapolation of $D(T)$ to $T = 0$ K for $Fe_3Sn_2$ using the power law fit is likely over estimated due to its validity only near $T_C$. The value of $D(T = 0$ K$) = 231 \pm 7$ meV $Å^2$ using the Dyson formalism is a better estimate of the ground state spin stiffness, although the limited temperature range accessible in this study still results in a large extrapolation window.

All ferromagnets have a gap due to the magnetic dipole-dipole interaction between different atoms [40]. This gap is typically small and often out of the range of resolution for inelastic neutron scattering experiments. The dipole-dipole interaction or resolution/instrumental alignment effects are both probable reasons for the observation of a small gap in this study (on the order of 0.06 meV to 0.09 meV), which is quite small compared to the exchange energy rendering $Fe_3Sn_2$ an isotropic ferromagnet to an excellent approximation. However, the magnetic anisotropy energy due to magnetocrystalline anisotropy was recently calculated to be close to our gap value, at 0.037 meV per Fe atom for the ground state when the easy-axis and spins are oriented within the kagome plane [41]. The ground state for which this value was calculated, though, is in a different temperature regime and spin configuration than that of the present experiment, so it is unclear whether the gap observed is solely due to magnetic anisotropy energy coming from dipolar interactions and/or magnetocrystalline effects.

We now discuss the meaning surrounding the term "anisotropy" in our discussion. Previous studies have cited the uniaxial anisotropy in $Fe_3Sn_2$ as one of the necessary ingredients for the formation of the topologically protected skyrmionic bubbles [10], and many of the centrosymmetric skyrmion systems discovered thus far are well-known to be a result of competition between frustrated magnetic exchange and spin anisotropy [21,22]. Unsurprisingly, measurements of the anisotropy energy density, $K_u$, have therefore been published [11,14,15] and show that the onset of a magnetic anisotropy precedes the temperatures at which the skyrmion bubbles are found. However, anisotropy can range from preferred orientation of a spin—which all ordered crystalline magnets have—to an appreciable energy required to pull spins away from a preferred direction. As a soft ferromagnet, $Fe_3Sn_2$ falls into the former category and can be considered an isotropic ferromagnet in accordance with our results. This is in contrast to the large anisotropies required for permanent magnet devices for magnetostatic energy storage [42]. In fact, one of the appealing properties of skyrmion-based devices may be that the weak anisotropy requirements open up the field for potential skyrmion candidate materials, especially when considering inducing small anisotropies into materials via doping is quite common.

An example of a system that internally tunes its anisotropy is $Nd_2Fe_{14}B$, a hard uniaxial ferromagnet used in permanent magnet applications but also exhibits a spin-reorientation transition such as that in $Fe_3Sn_2$. It was found that the rotating spins act to tune the overall anisotropy in the system [43], although in contrast with $Fe_3Sn_2$, the anisotropy is due to the lanthanide crystal field effect. It is also instructive to recall that spin anisotropy is not required for skyrmion formation in inversion symmetric systems [19,20], although these theories have not been specifically applied yet to $Fe_3Sn_2$. Another metallic breathing kagome lattice to host a skyrmion spin texture is $Gd_3Ru_4Al_{12}$ [44]. In contrast to $Fe_3Sn_2$, the ordered magnetic state is antiferromagnetic and a weak anisotropy is of the easy-plane type. Future work on either of these kagome materials to directly probe the anisotropy gap in proximity to the skyrmion phases would be of interest to explore the role of the anisotropy versus magnetic exchange frustration. Inelastic neutron scattering can be used to achieve this below the temperatures accessible in the work presented here but would require a comparable mass of co-aligned single crystals and sub-meV instrumental resolution in a wide-angle scattering experiment.

**Author Contributions:** Material synthesis, N.J.G. and N.B.; Magnetization measurements, D.P.; neutron experiment design, J.W.L. and R.L.D.; neutron experiment analysis, J.W.L. and R.L.D.; writing—original draft preparation, R.L.D.; writing—review and editing, J.W.L., R.L.D., D.P., and N.J.G. All authors have read and agreed to the published version of the manuscript.

**Funding:** Synthesis and characterization work (N.J.G.) were supported by the U.S. Department of Energy, Office of Science, Basic Energy Sciences, Materials Science and Engineering Division. Work in the Materials Science Division at Argonne National Laboratory was supported by the U.S. Department of Energy, Office of Science, Basic Energy Sciences, Materials Science and Engineering Division.

**Institutional Review Board Statement:** Not applicable.

**Informed Consent Statement:** Not applicable.

**Data Availability Statement:** Data is available upon request to the corresponding author.

**Conflicts of Interest:** The identification of any commercial product or trade name does not imply endorsement or recommendation by the National Institute of Standards and Technology. The authors declare no conflict of interest. The funders had no role in the design of the study; in the collection, analyses, or interpretation of data; in the writing of the manuscript, or in the decision to publish the results.

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
