# Peer review of "Isotropic Nature of the Metallic Kagome Ferromagnet Fe3Sn2 at High Temperatures"

_crystals, doi:10.3390/cryst11030307_

Round 1

Reviewer 1 Report

The paper by R. Dally et al. considers magnetic properties of Fe3Sn2 compound at increased temperatures. The authors have focused on the results of small-angle inelastic neutron scattering measurements, which have indicated that for the considered material the magnetic anisotropy in the temperature range from 480 K up to Tc \approx 660 K may be neglected. The examined Fe3Sn2 compound has recently been the subject of intensive research since it may be considered as a prototypical antiferromagnetic kagome lattice system (Kang et al. Nature Materials 19 (2020) 163-169), Ghimire and Mazin, Nature Materials 19 (2020) 1301-138).

The paper is well written and it deserves publication “as it is”. The study of magnetic interactions in complex structures like the one examined in this paper has been in the spotlight of physicists and the paper might attract attention of other researchers. The authors are experienced scientists  coming from top US research centers and they have published a remarkable number of papers on the very compound and on similar subjects in the most influential journals. This may be considered as a warranty of the correctness of their work.

Inquiry to the authors: please explain the choice of the lower temperature used in the experiments. It seems that for the value 480 K the discrepancies between the Dyson law and the mean field theory become negligible (Fig. 3). Was this one of the premises for choosing this temperature?

Author Response

We thank the Reviewer for their thorough reading and comments. To address the inquiry they have: "please explain the choice of the lower temperature used in the experiments. It seems that for the value 480 K the discrepancies between the Dyson law and the mean field theory become negligible (Fig. 3). Was this one of the premises for choosing this temperature?"

Below 480 K, the spin waves move out of our measurement window because (a) the spin stiffness increases and (b) they narrow in energy due to the decrease of damping from magnon-magnon scattering. So the choice of 480 K as a lower bound was purely because the measurement was not possible on the instrument below this temperature. To be clear - we could still the "tails" of the spin waves well below this temperature, but couldn't collect enough data to properly analyze and fit it. The Dyson law is not applicable close to T_C, which is why we chose to use the lower four temperatures when fitting. This choice was somewhat arbitrary, as there isn't a hard temperature limit for the law. As we stated in the manuscript, the D(T=0) value extracted from the Dyson law is only an estimate. In an ideal world, where we could obtain data at lower temperatures, the fit would provide a more accurate estimate.   

Reviewer 2 Report

Neutron scattering measurements show that Fe3Sn2 is an isotropic ferromagnet down to 480K but at lower temperatures the neutron measurement window is lost. At least, the authors could have extended the low-temperature magnetic characterization by magnetometry. For example, the authors could demonstrate the soft and isotropic character of their Fe3Sn2 samples by measuring hysteresis curves with the field applied in different directions and at different temperatures in the ranges: T<150 K, 150 <T<250 K  and 300 <T < Tc.

Other suggestions:

Line 8: The term lattice anisotropy is often used among magnetics researchers to refer to magnetocrystalline anisotropy. Please clarify the expression lattice anisotropy so that there is no confusion.

Line 10: Authors must refer a specific value next to this expression “Our results reveal that at elevated temperatures…”

Figure 1d: Were the hysteresis loops measured with a magnetic field applied parallel or perpendicular to the ab plane? Give this information in the text.

Would there be coercivity and easy/hard axes if the magnetic field had been applied in the direction perpendicular to the measurement showed in Fig 1d? And in other directions in the ab plane?

The discussion is somewhat diffuse and general. Many of the aspects discussed could be included in the introduction because they refer to the state of the art on the subject. The authors could be a little more concrete when discussing their results.

The conclusions of the work are missing. The authors should include them.
